# Simulation and Projection of Climate Extremes in China by a Set of Statistical Downscaled Data

**DOI:** 10.3390/ijerph19116398

**Published:** 2022-05-24

**Authors:** Linxiao Wei, Lyuliu Liu, Cheng Jing, Yao Wu, Xiaoge Xin, Baogang Yang, Hongyu Tang, Yonghua Li, Yong Wang, Tianyu Zhang, Fen Zhang

**Affiliations:** 1Chongqing Climate Center, Chongqing 401147, China; quiet7@126.com (L.W.); cuit_sky@163.com (Y.W.); yangbaogang0418@126.com (B.Y.); tanghy99@126.com (H.T.); lyhcq@163.com (Y.L.); wngyng023@126.com (Y.W.); zhangtianyu821227@hotmail.com (T.Z.); zhangfen@mail.bnu.edu.cn (F.Z.); 2National Climate Center of China Meteorological Administration (CMA), Beijing 100081, China; 3School of Geographical Science, Nanjing University of Information Science & Technology, Nanjing 210044, China; jingc1992@163.com; 4Center for Earth System Modeling and Prediction of CMA (CEMC), Beijing 100081, China; xinxg@cma.gov.cn; 5State Key Laboratory of Severe Weather (LaSW), Beijing 100081, China

**Keywords:** evaluation, projection, CMIP6, downscaling, extreme climate

## Abstract

This study assesses present-day extreme climate changes over China by using a set of phase 6 of the Coupled Model Intercomparison Project (CMIP6) statistical downscaled data and raw models outputs. The downscaled data is produced by the adapted spatial disaggregation and equal distance cumulative distribution function (EDCDF) method at the resolution of 0.25° × 0.25° for the present day (1961–2014) and the future period (2015–2100) under the Shared Socioeconomic Path-way (SSP) 2-4.5 than SSP5-8.5 emission scenario. The results show that the downscaling method improves the spatial distributions of extreme climate events in China with higher spatial pattern correlations, Taylor Skill Scores and closer magnitudes no matter single model or multi model ensemble (MME). In the future projections, large inter-model variability between the downscaled models still exists, particular for maximum consecutive 5-day precipitation (RX5). The downscaled MME projects that total precipitation (PTOT) and RX5, will increase with time, especially for the northwest China. The projected heavy precipitation days (R20) also increase in the future. The region of significant increase in R20 locates in the south of river Yangtze. Maxi-mum annual temperature (TXX) and percentage of warm days (TX90p) are projected to increase across the whole country with larger magnitude over the west China. Projected changes of minimum annual temperature (TNN) over the northeastern China is the most significant area. The higher of the emission scenario, the more significant of extreme climates. This reveals that the spatial distribution of extreme climate events will become more uneven in the future.

## 1. Introduction

The greenhouse gas emissions have caused the global average surface temperature to increase by about 1 °C in the first two decades of the 21st century (2001–2020) compared to the pre-industrial level. Climate change is already affecting climate extremes and has seriously threatened human systems and terrestrial and marine ecosystems [1,2,3,4,5,6,7,8]. Extreme climate events in China have increased significantly in recent decades [9,10], such as the severe drought in southwest China from 2009 to early 2010, the high-temperature heatwave in south China in 2013, the super-strong precipitations over the Yangtze River in the prolonged Meiyu Period of 2020 that caused heavy floods in the middle and lower reaches of the river Yangtze [11,12], and the torrential rain in Henan in July 2021, with the daily precipitation breaking a record at 10 meteorological stations [13]. The losses caused by meteorological disasters account for about 71% of the total losses of all kinds of natural disasters in China [14].

Compared with the 1 °C warming level, extreme climate events, such as glaciers and snowmelt, and sea-level rise, will be further aggravated at a 1.5 °C warming level between 2030 and 2052 (IPCC, 2018). More unevenly spatial and temporal distribution of global land precipitation and a faster increase in the frequency and intensity of extreme precipitation and drought events were expected in the 21st century. Climate change has become a global challenge. It is of great importance to project changes in extreme climatic events as a means of mitigating the social and economic impacts of climate change. China is located in the East Asian monsoon region and is more vulnerable to extreme precipitation events due to significant thermal differences and complex topography. Additionally, it will be threatened by a fast increase in extreme events because the rarer the extreme climate events are, the faster the frequency will increase [15,16].

Global Climate Models (GCMs) are the most popular tool to project future climate change. Their simulation biases are large at local and regional scales, and there is greater uncertainty, especially in China as a monsoon region [15,16]. However, policymakers often need simulation with a fine resolution of 1–50 km or even finer. Dynamical and statistical downscaling methods are usually used to bridge the gap. Each one has advantages and limitations. Dynamic downscaling can better simulate finer-scale physical processes such as effects of topography, land-sea distribution, and land surface processes, so it can improve the simulation of temperature and precipitation at the regional scale relative to the global model [17,18,19,20,21]. However, it requires high computing resources and is time-consuming. Systematic biases of the GCM will be carried into the regional climate model. While statistical downscaling is time-saving and requires fewer computing resources. A long series of fine climate simulations can be established quickly by correcting the climatological mean error. However, it lacks a description of the physical mechanism. Different downscaled datasets have been established aiming for various purposes and on various basis of computing resources. For example, Jiang et al. [22] applied the equal distance cumulative distribution function (EDCDF) method to statistically downscale 12 GCMs of phase 5 of the Coupled Model Intercomparison Project (CMIP5) from the World Climate Research and established a daily climate data set of temperature, precipitation, relative humidity, wind speed, and radiation of China with a resolution of 0.5° × 0.5° to project flood losses in China under different temperature rise scenarios. NASA performed statistical downscaling and bias correction based on 21 GCMs models of CMIP5 to build a global daily dataset of daily maximum and minimum temperature and daily precipitation (https://www.nccs.nasa.gov/services/data-collections/land-based-products/nex-gddp, accessed on 20 May 2022) with a resolution of 0.25° × 0.25°, which significantly improved simulation capability at regional and local scales and reduced the uncertainty of the estimated results by narrowing differences among models [23,24,25,26] compared the capabilities of four statistical downscaling methods for extreme climate events in China and found that the extreme precipitation index simulated by the bias correction and spatial downscaling (BCSD) method was obviously better than the other three methods but had a large bias in the simulation of extreme temperature events.

Although the resolution and physical processes of the global climate models of the CMIP6 have been improved, downscaling is still necessary for the projection and impact assessment of climate change at a regional scale. A dataset of daily maximum and minimum temperatures and daily precipitation in China from 1961 to 2100 with a 0.25° × 0.25° resolution was established by the National Climate Center, China Meteorological Administration, by spatial downscaling and bias correcting based on 13 GCMs of CMIP6. Its application in the Yellow River Basin shows that this dataset outperforms CMIP6 models in terms of the climatological pattern of annual average temperature and annual precipitation over the Yellow River Basin [27]. This research here aims to evaluate the capability of this dataset in simulating extreme climate events over China and to project the changes of these events over China in the 21st century under different emission scenarios. This dataset can provide more reliable and refined projections and improve the quality of decision-making services in response to climate change. It also provides scientific references for the ecological environment, ecological security, socio-economic, and development planning in the region.

This paper analyzes the observation, raw CMIP6 model outputs, and downscaled CMIP6 data to answer the following questions: (1) Is there a significant improvement in the results of the downscaled CMIP6 data compared to the raw outputs? (2) What will be the characteristics of future extreme climate in China under different emission scenarios by using the downscaled data?

## 2. Models, Data and Methods

### 2.1. Data

Observational climate data are daily precipitation, daily maximum, and minimum temperature series with a resolution of 0.25° × 0.25° by interpolating 2400 meteorological observation stations in China from 1961 to 2014 with the method of “anomaly approach” [28,29].

### 2.2. Models

Climate model data include raw outputs and corresponding downscaled data of 13 climate models (Table 1) of the CMIP6 in the historical period of 1961–2014, and in the period of 2015–2100 under SSP2-4.5 and SSP5-8.5 emission scenarios. The Shared Socioeconomic Pathway (SSP) 2-4.5 represents the intermediate emission scenario, this scenario stabilizes the radiative forcing at 4.5 W/m^2^ in year 2100. SSP5-8.5 represents the very high emission scenario, this scenario stabilizes the radiative forcing at 8.5 W/m^2^ in year 2100. SSP2-4.5 represents the most likely future emissions scenario, and SSP5-8.5 represents the most extreme scenario with no control of fossil fuel emissions. The comparison under these two scenarios will help to highlight the importance of carbon neutrality.

### 2.3. Methods

#### 2.3.1. Statiscal Downsacling Method

The downscaled data was produced using adapted spatial disaggregation downscaling method [22,27] based on the previous method [30], and bias-correcting method of equidistant cumulative distribution functions (EDCDF) [31]. The method is superior to the traditional method [32,33].

(1) Firstly, the mean observed climate variables during 1961–2014 with 0.25° × 0.25° resolution were interpolated to General Circulation Model (GCM) coarse resolution by bilinear interpolating for each month separately. Secondly, coarse resolution anomaly fields were calculated as the difference between the observation and climate model outputs. Thirdly, the anomaly fields were interpolated to 0.25° × 0.25° resolution by bilinear interpolating, and then added to the observation to obtain the downscaled GCM results [34]. It was assumed that the topographic and climatic features would remain unchanged after spatial disaggregation and in future periods.

(2) Statistical downscaled General Circulation Model (GCM) simulations were bias-corrected with equidistant cumulative distribution functions (EDCDF) method [31], which assumes that the difference between the observation and the simulation in the training period is same in any selected simulation period for a given percentile. Temperature is fitted based on normal distribution, while precipitation is based on mixed gamma distribution. Finally, original GCM datasets were bias-corrected by adding the difference. The equation of the method can be written as follows:(1)$$C_{v,cor}=C_{v}+F_{ot}^{-1}\left(F_{mc}\left(C_{v}\right)\right)-F_{mt}^{-1}\left(F_{mc}\left(C_{v}\right)\right)$$

In the equation, *F_mc_* is the cumulative distribution function (CDF) and $F_{ot}^{-1}$ and $F_{mt}^{-1}$ are the inverse CDF; *C_v_* is the climate variable value (*ot* denotes observations in the training period; *mt* denotes model outputs in the training period (1961–2014); *mc* denotes model outputs in a correction period).

Using the downscaling and bias-correcting method, the downscaled dataset was produced to a common 0.25° × 0.25° grid including daily maximum temperature, daily minimum temperature, and daily precipitation from 13 GCMs.

#### 2.3.2. Evaluation Method

The capability of downscaled CMIP6 GCMs was evaluated by comparing with the raw outputs of GCMs from climatological mean pattern and temporal trends of six climate extreme indices (Table 2) (https://www.Climdex.org/learn/Indices/#index-GSL, accessed on 20 May 2022) over China, considering that these extreme events may have serious socio-economic impacts on China. PTOT is the total annual precipitation. RX5, TXX, and TNN represent the intensity of extreme events, while R20 and TX90P represent the frequency of extreme events.

The climatological mean was calculated at each grid from 1961 to 2014 for observation, downscaled, and raw outputs of GCM. Then the simulation capability was demonstrated by spatial graph and metric analysis. The spatial graph shows the spatial distribution of multi-model ensemble average, which was calculated with the equal-weight arithmetic average method [35]. The pattern correlation between the simulated and observed fields (*R*), relative standard deviation ($\hat{\sigma }$), and Taylor Skill (*TS*) score [36] were used to quantitatively evaluate the simulation capability of the two sets of data for each extreme climate index. *TS* score can directly reflect the performance of the two sets of data.
$$TS=\frac{4{\left(1+R\right)}^{2}}{{\left(\frac{\sigma_{sm}}{\sigma_{so}}+\frac{\sigma_{so}}{\sigma_{sm}}\right)}^{2}+{\left(1+R_{0}\right)}^{2}}$$
where, *R* is the pattern correlation coefficient between the simulation and observation, and $R_{0}$ is correlation coefficient attainable (0.999 used here). $\sigma_{sm}$ is the standard deviation of the simulated spatial field, and $\sigma_{so}$ is the standard deviation of observed spatial field. If the simulation reproduces the observed spatial pattern perfectly, TS would equal to 1.

## 3. Results

### 3.1. Spatial Evaluation

Figure 1 shows the spatial distribution of extreme precipitation indices for observation, multi-model-ensemble (MME) of CMIP6, and MME of the downscaled field during 1961–2014. The observed PTOT, RX5, and R20 all show a spatial distribution characteristic of gradually increasing from northwest to southeast. PTOT gradually increases from less than 200 mm in northwestern China to more than 2000 mm in the southeast coastal areas. RX5 precipitation gradually increases from less than 20 mm in northwestern China to more than 200 mm in the southeast coastal areas. R20 gradually decreases from more than 20 days to less than 5 days in northwestern China.

Both the CMIP6 MME and the downscaled MME can well reproduce the spatial distribution pattern. Compared with CMIP6, the downscaling method effectively reduces the simulated biases. The PTOT and RX5 simulated by CMIP6 have an obvious positive bias in southeastern China, while downscaling technology reduces these biases well. The observed high-value centers of R20 appear in Guangdong and Jiangxi, with a period from 25 to 30 days, while the simulation of CMIP6 has a large high-value area. After downscaling, the bias is reduced.

Table 3 lists the evaluation metrics of extreme indices. R of PTOT, RX5, and R20 for CMIP6 MME are 0.86, 0.88, and 0.79, respectively, and $\hat{\sigma }$ is 1.30, 1.34, and 1.24, and the *TS* score is 0.81, 0.81, and 0.76, respectively. The values of R and TS are more than 0.99 after downscaling. These results indicate that the spatial pattern is improved by the downscaling method. The same is true for most individual models, especially for BCC-CSM2-MR. The TS of RX5 is 0.93 after downscaling, while it is only 0.42 for the raw output of the model.

Figure 2 shows the spatial distribution of extreme temperature indices for observation, CMIP6 MME, and downscaled MME during 1961–2014. The observed high-value center of TXX appears in the eastern part of Xinjiang, which is above 40 °C. The CMIP6 model and statistical downscaling both reproduce this spatial distribution pattern. However, the simulated values of CMIP6 are higher in southern Xinjiang and the southeast of northern China, and lower in the southeast coastal areas, while the downscaled results are very close to the observed ones. The observed low-value areas of TX90P are concentrated in central and eastern Henan, western Shandong, and southern Hebei. Both the CMIP6 model and downscaling results underestimate TX90P in most central and eastern regions of China. The TNN of the three sets of data displays a characteristic of gradually decreasing from south to north. After downscaling, the cold bias in the upper river Yangtze is effectively reduced, which is very close to the observation.

Table 3 shows that *R* of TXX, TNN and TX90P for CMIP6 MME are 0.94, 0.95, and 0.47, $\hat{\sigma }$ are 0.92, 1.79, and 2.02, and TS are 0.94, 0.69, and 0.34, respectively. *R*, $\hat{\sigma }$ and *TS* are close to 1 for the downscaled dataset. This suggests that the downscaling method improves the simulation of spatial patterns of extreme temperature indices.

All these analyses show that this set of downscaling data can better reproduce the spatial distribution patterns of the three extreme precipitation events in China, namely, PTOT, RX5, R20, TXX, and TNN, no matter from the ensemble average or single models, which is consistent with the conclusion of Yang et al. [25,26].

### 3.2. Temporal Evaluation

Table 4 lists the trend coefficients of regional averages of the indices over China from 1961 to 2014 for observation, CMIP6, and downscaled MME. The regional mean PTOT of observation, CMIP6 MME, and downscaled MME changed at the rates of 0.6%, 0.09%, and −0.2% per decade, respectively. RX5 increased at a rate of 0.3%, 0.7%, and 0.6% per decade, and R20 varied negligibly. The increase rate of RX5 simulated by CMIP6 and downscaling is larger than the observation, but the downscaling is closer to the observation. However, the downscaling method fails to simulate the weak increasing trend of PTOT. For extreme temperature indices, both the CMIP6 and downscaling reproduce the increasing trends of TX90P and TNN, with a slower rate for TNN than observation, but fail to reproduce the trends of TXX. Compared with CMIP6 MME, the rate of TX90P is closer to the observation after downscaling. The temporal trends of precipitation at a regional scale are difficult to simulate, which was found in previous studies [37,38,39]. Figure 3 also shows that the downscaled MME is unable to improve the temporal variation of extr0eme climate indices.

## 4. Future Projection Based on Downscaled CMIP6 Multi-Model Ensemble

### 4.1. Temporal Evoluation

Figure 4 shows the changes in China’s regional averaged extreme climate indices from 2021 to 2100 based on the downscaled data under the SSP2-4.5 and SSP5-8.5 scenarios. Under these two scenarios, the extreme precipitation and temperature events in China will increase and increase faster under SSP5-8.5 than under SSP2-4.5. By the end of the 21st century, the averaged PTOT, RX5, R20, TXX, TX90P, and TNN will increase by 15%, 20%, 1.4 days, 3 °C, 30%, and 4 °C, respectively, under the SSP2-4.5 scenario, and will increase by a faster rate of 30%, 30% and 2.2 days, 6.4 °C, 57% and 8 °C, respectively, under the SSP5-8.5 scenario. The amplitudes of change among different models are significantly different. The uncertainty range caused by climate models increases gradually with the extension of projection time, and the uncertainty under SSP5-8.5 is greater than under SSP2-4.5.

### 4.2. Spatial Evolution

The spatial changes of extreme climate projections were calculated for the anomaly of near-term (2021–2040), mid-term (2041–2060), and long-term (2081–2100) periods relative to the base period (1995–2014) under the two scenarios. PTOT and RX5 were expressed by anomaly percentage, and other extreme climate indices were expressed by anomaly value.

Figure 5 shows the spatial patterns of projected changes in PTOT, RX5, and R20 under the SSP2-4.5 scenario. Overall, PTOT over China is projected to increase, but the magnitude of the increase varies greatly from region to region. It will increase by more than 5% in most parts of China, especially in southern Xinjiang and western Tibet, where the increase will exceed 40% in the near-term period. The high-value center of PTOT is projected to expand to the north with time in the mid- and long-term period. The projected changes in RX5 are similar to PTOT, where the maximum center is also located in northwestern China, with an increase of approximately 40%. These changes will result in a greater increase in northern China than in southern China for PTOT and RX5. R20 is projected to generally increase over the whole country, it would increase approximately by 1 day in the near-term period. It would be up to 2 days and 3–4 days over southern China in the mid-term and long-term periods. The above results are consistent with the conclusions of Xu et al. [40] and Zhu et al. [41].

Under the SSP5-8.5 scenario (Figure 6), the projected changes in extreme precipitation indices are similar to those under SSP2-4.5. However, the increase is larger than that under SSP2-4.5. PTOT will increase by more than 60% in the northwest of China. The projected distribution pattern of RX5 is still similar to PTOT, with a maximum center exceeding 60% located over the west of China. R20 is projected to increase by more than 5 days in southeastern China. On the whole, the increase in PTOT and RX5 would be stronger in northern and western China than in the rest of the parts of China. The south of the river Yangtze is the region with the most significant increase in R20. The above findings are also confirmed by Chen et al. [42] and Li et al. [10], who used statistical and dynamical downscaling methods to explore the future changes of extreme climate events in China. The distribution map of the extreme precipitation is similar to the above results.

Figure 7 shows the spatial changes of extreme temperature indices under SSP2-4.5. TXX is projected to unevenly increase over the whole country. TXX in northwest China would warm the most. The projected temperature increase of this region varies from 1 °C to 4 °C over time. TX90P is projected to increase in most parts of China. The increase in west China is more remarkable than in other areas, which will increase by over 35% in the long-term period. TNN is projected to increase intensely over northeastern China and with the value exceeding 4 °C in the long-term period.

Under the SSP5-8.5 scenario (Figure 8), the projected changes are similar to those under SSP2-4.5 except for the magnitude of the increase. The increase in TXX, TX90P, and TNN is also larger with time. The projected increase in TXX is unevenly distributed. TXX over northwestern China would be warm by the most with an increase of approximately from 1 °C to 7 °C. The increase of TX90P with the maximum center value located in Xizang exceeded 60% in the long-term period. The evolution of TNN in the near-term period under SSP5-8.5 is basically the same as that under SSP2-4.5. Northeastern China is the region where the increase in TNN is most pronounced in the mid- and long-term periods. TNN over this region would warm by 7 °C. This indicates that the higher the SSP scenario, the stronger the projected warming. The above analysis is consistent with the conclusion of Yang et al., 2018 and Zhu et al., 2021, who used the outputs of the statistical downscaled projection of the CMIP5 and 20 models from CMIP6. Ge et al. [43] used a regional climate model (RCM) to project the future hot extremes. The dynamical downscaling method also indicates more intense and frequent hot extremes will occur due to global warming.

## 5. Discussion

This finding is significant because it first used spatial disaggregation downscaling and EDCDF method to downscale 13 CMIP6 models. The projections have a very high resolution of 25 km. So, the results of this study can provide a better understanding of the regional variability of extreme climate events and thus provide policy makers with corresponding climate change mitigation and adaptation strategies. Wang and Chen [44] found the EDCDF method will result in negative precipitation, they provide a feasible alternative called equiratio CDFm to solve this problem. We will use the method to downscale the climate model in next paper. In order to understand the physical processes of future changes in extreme climate events, the Regional Climate Model (RCM) should be used in the future to conduct sensitivity tests of different physical processes to deepen the understanding of the causes of extreme climate events. Then the future changes of extreme climates can be more reliably projected. The internal variability, inter-model spread and uncertainty in future emissions are the primary sources of the uncertainties in future projections. Therefore, it is worthwhile to try to combine the dynamic downscaling and statistical downscaling methods to improve the reliability of climate prediction. We will explore this topic in the future.

## 6. Conclusions

In this study, the EDCDF statistical downscaling method driven by 13 CMIP6 models was evaluated via a comparison with observations in China for the historical period (1961–2014), as applied to raw GCMs from CMIP6. The spatial pattern correlation, relative standard deviation, and *TS* score are used to quantitatively evaluate the spatial patterns simulated by models. Then the downscaled MME has been applied to investigate the future changes in extreme climates in the 21st century under SSP2-4.5 and SSP5-8.5. 

Statistical downscaling models generally simulate more reasonable spatial distribution patterns of PTOT in China by successfully eliminating the artificial precipitation maximum area over the south of the Tibetan Plateau and southern China. The statistical MME also improves the capability of reproducing the spatial pattern of RX5, R20, TXX, and TNN. However, this method still has difficulties in simulating the distribution of TX90P. The spatial correlation and *TS* scores of downscaled MME for PTOT, RX5, R20, TXX, and TNN are more than 0.99, which is far better than the MME of raw models. The spatial correlation and *TS* score of the downscaled TX90P also improved; the values are between 0.52 and 0.44, respectively, compared to between 0.47 and 0.34 in the original model. In addition, the simulation capability of individual models has been greatly improved after downscaling. In all, the statistical method displays obvious advantages in simulating the spatial patterns over its driving GCMs. The downscaled MME fails to improve the simulation of interannual variation in the extreme indices.

We further explore the future changes in the six extreme climates under the two scenarios in the 21st century. PTOT and RX5 increased by more than 30% at the end of the 21st century under SSP5-8.5. Northwestern China is the area with the most significant changes, with an increase of the two indices of more than 60%. The projected increase in PTOT and RX5 in north China is more significant than in south China. R20 is projected to increase by 2.2 days at the end of the 21st century under SSP5-8.5. Southern China would increase the most, with a value of more than 5 days. Large inter-model variability between the downscaled models still exists in extreme climate events, particularly for maximum consecutive 5-day precipitation (RX5). TXX would warm by 6.4 °C and TNN would warm by 8 °C. Warming would be of different strengths in different regions. Northwest China and northeast China are the regions with the most significant warming, with a value of more than 7 °C under SSP5.8.5 for TXX and TNN, respectively. The Tibet Plateau region is the most significant region for the increase of TX90P under SSP5-8.5, with a value of more than 60%. The whole country will increase by 57% at the end of the 21st century under SSP5-8.5. The downscaled MME projects that China will become warmer and wetter in the future. It will be affected by more intensive and frequent extreme climate events. The increase will be larger as time extends. The higher the SSP scenario, the stronger the extreme climates become. The projected increase in the six extreme climate events will be unevenly distributed.

## Figures and Tables

**Figure 1 ijerph-19-06398-f001:**
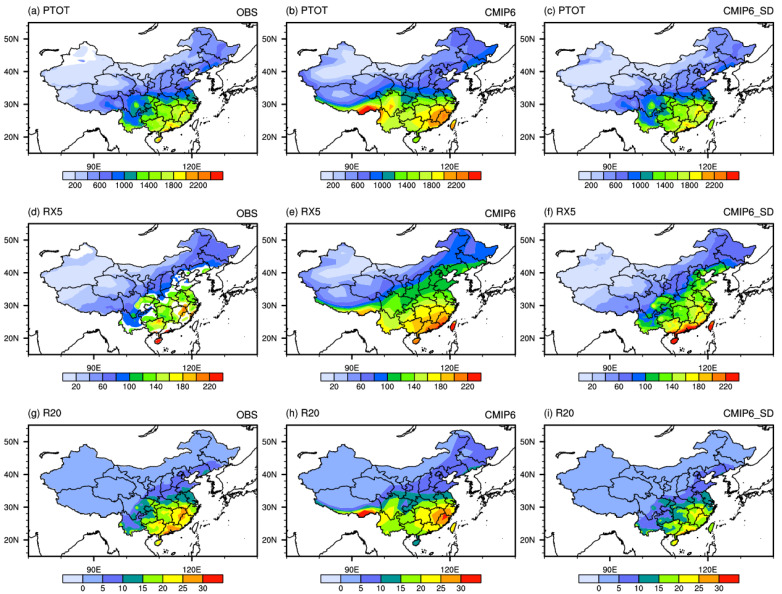
Spatial patterns of extreme precipitation indices (PTOT and RX5, units: mm; R20, units: days) in China (1961–2014) for observation (**left** panel), CMIP6 multi-model-ensemble (MME) (**middle** panel), and MME of downscaled dataset (**right** panel).

**Figure 2 ijerph-19-06398-f002:**
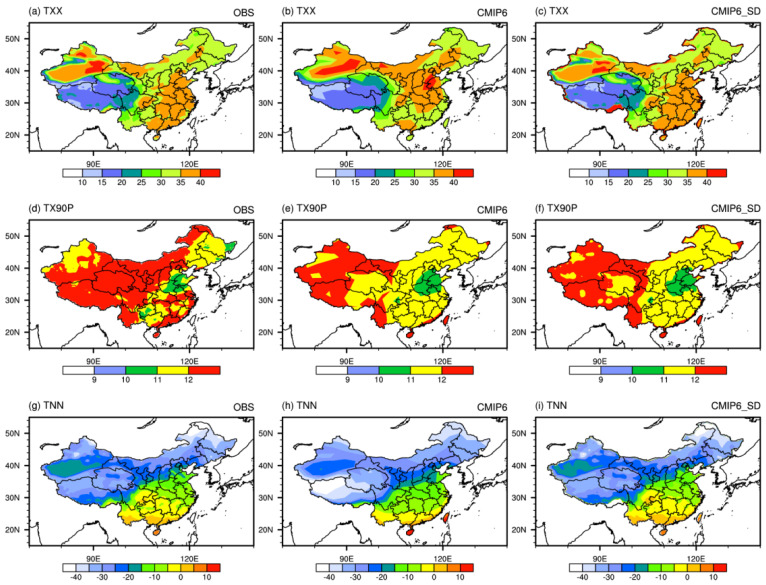
Spatial patterns of extreme temperature indices (TXX and TNN, units: °C; TX90P, units: %) in China (1961–2014) for observation (**left** panel), CMIP6 MME (**middle** panel), and MME of downscaled dataset (**right** panel).

**Figure 3 ijerph-19-06398-f003:**
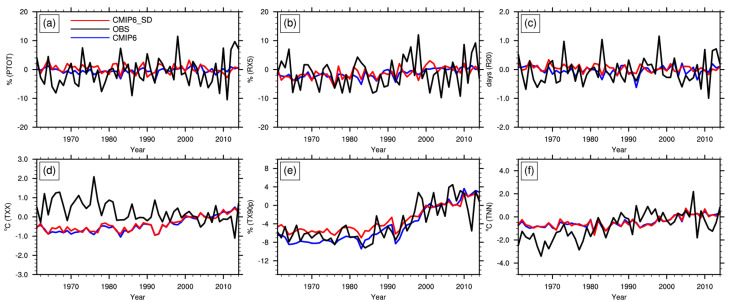
Extreme climate indices of PTOT (**a**), RX5 (**b**), R20 (**c**), TXX (**d**), TX90P (**e**) and TNN (**f**) over China during 1961–2014 for observation, CMIP6 MME and MME of downscaled dataset (black lines indicate observations, blue lines indicate CMIP6, and red lines indicate MME of downscaled dataset).

**Figure 4 ijerph-19-06398-f004:**
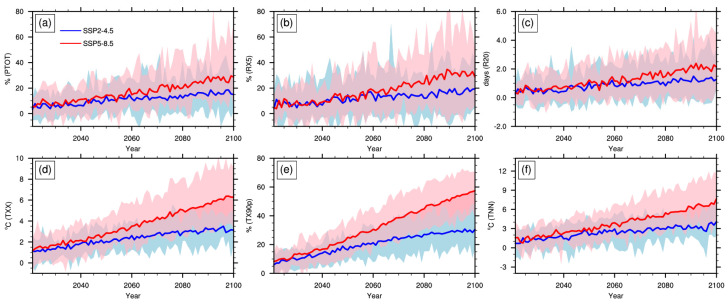
Projected changes in extreme climate indices of PTOT (**a**), RX5 (**b**), R20 (**c**), TXX (**d**), TX90P (**e**) and TNN (**f**) over China from 2021 to 2100 relative to baseline (1995–2014) based on downscaled dataset under scenarios SSP2-4.5 (blue) and SSP5-8.5 (pink) (shaded area represents for the range within maximum and minimum of the 13 models; lines are the multi-model ensemble means).

**Figure 5 ijerph-19-06398-f005:**
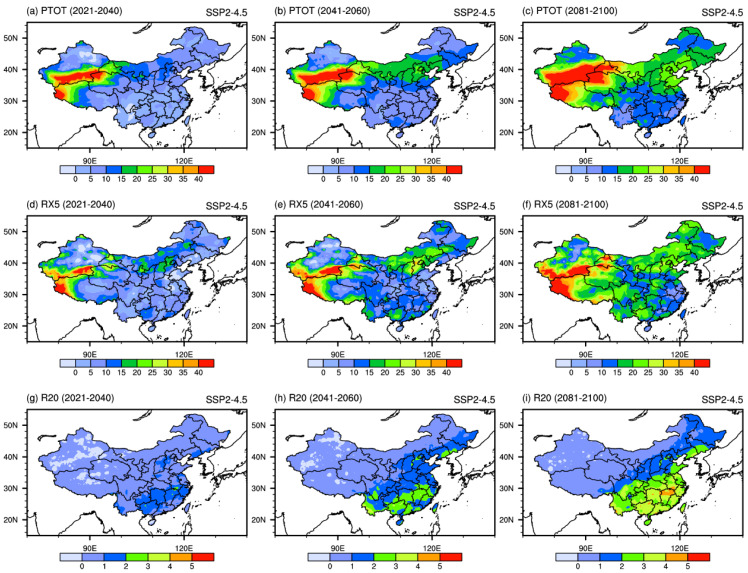
Spatial patterns of extreme precipitation indices (units: %) projected by the downscaled CMIP6 MME under SSP2-4.5 scenario in the 21st century.

**Figure 6 ijerph-19-06398-f006:**
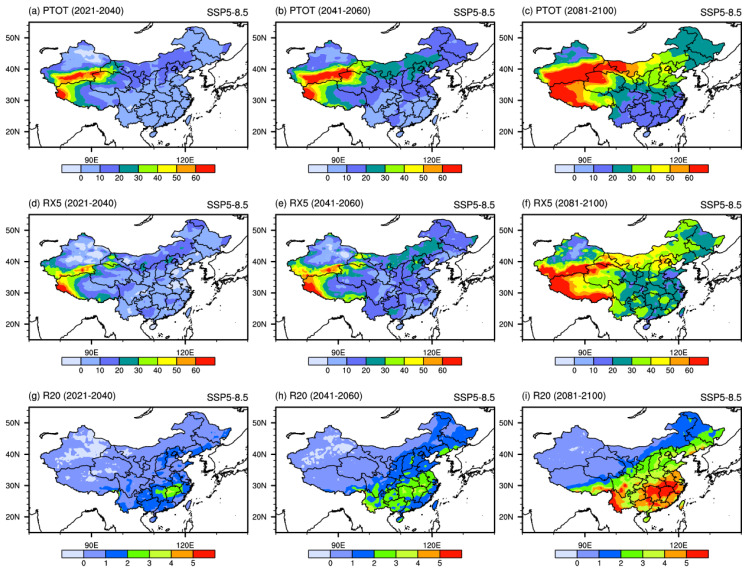
Spatial patterns of extreme precipitation indices (units: %) projected by the downscaled CMIP6 MME under SSP5-8.5 scenario in the 21st century.

**Figure 7 ijerph-19-06398-f007:**
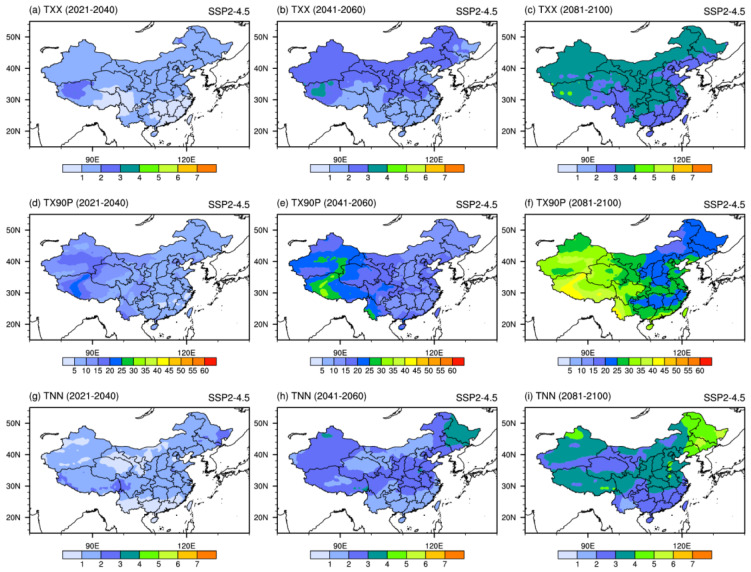
Spatial patterns of extreme temperature indices (TXX, TNN, units: °C; TX90P, units: %) projected by the downscaled CMIP6 MME under SSP2-4.5 scenario in the 21st century.

**Figure 8 ijerph-19-06398-f008:**
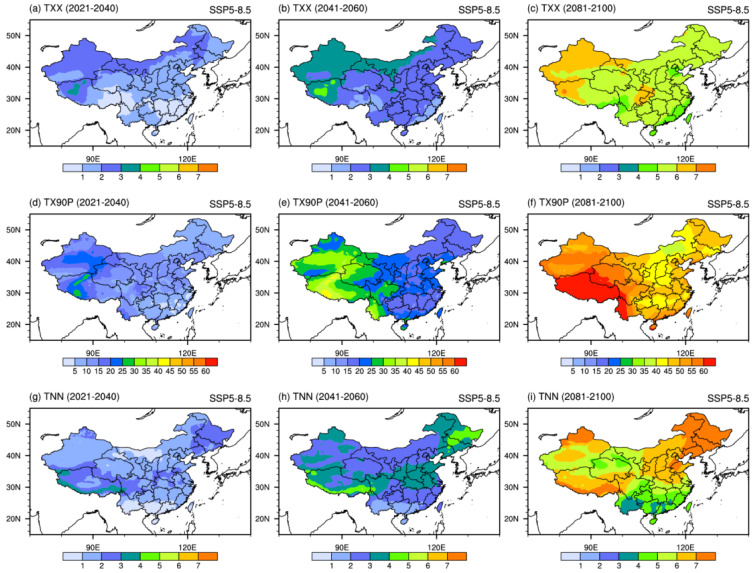
Spatial patterns of extreme temperature indices (TXX, TNN, units: °C; TX90P, units: %) projected by the downscaled CMIP6 MME under SSP5-8.5 scenario in the 21st century.

**Table 1 ijerph-19-06398-t001:** General inflormaiton of CMIP6 models used in this study.

ID	Model Name	Affiliated Country and Research Unit	Atmos. Lat/Lon Grid (°)
1	ACCESS-CM2	Commonwealth Scientific and Industrial Research Organisation, and Australian Research Council of Excellence for Climate System Science (Australia)	1.2° × 1.8°
2	ACCESS-ESM-1-5	Commonwealth Scientific and Industrial Research Organisation(Australia)	1.2° × 1.8°
3	BCC-CSM2-MR	Beijing Climate Center, China Meteorological Administration (China)	1.1° × 1.1°
4	CanESM5	Canadian Centre for Climate Modelling and Analysis (Canada)	2.8° × 2.8°
5	CNRM-CM6-1	Centre National de Recherches Météorologiques, Centre Européen de Recherche et de Formation Avancée en Calcul Scientifique (France)	1.4° × 1.4°
6	CNRM-ESM2-1	Centre National de Recherches Météorologiques, Centre Européen de Recherche et de Formation Avancée en Calcul Scientifique (France)	1.4° × 1.4°
7	HadGEM3-GC31-LL	Met Office Hadley Centre (United Kingdom)	1.3° × 1.9°
8	INM-CM4-8	Institute for Numerical Mathematics, Russian Academy of Science (Russia)	1.5° × 2.0°
9	INM-CM5-0	Institute for Numerical Mathematics, Russian Academy of Science (Russia)	1.5° × 1.5°
10	IPSL-CM6A-LR	Institut Pierre Simon Laplace (France)	1.3° × 2.5°
11	MIROC6	Japan Agency for Marine-Earth Science and Technology, Atmosphere and Ocean Research Institute (The University of Tokyo), National Institute for Environmental Studies, and RIKEN Center for Computational Science (Japan)	1.4° × 1.4°
12	MPI-ESM1-2-HR	Max Planck Institute for Meteorology (Germany)	0.9° × 0.9°
13	MRI-ESM2-0	Meteorological Research Institute (Japan)	1.1° × 1.1°

**Table 2 ijerph-19-06398-t002:** Defintions of extreme precipitation and temperature indices.

Name	Acronym	Definition	Unit
Total precipitation	PTOT	Annual total precipitation in wet days (daily precipitation larger than 1 mm)	mm
Maximum consecutive 5-day precipitation	RX5	Annual maximum consecutive 5-day precipitation	mm
Very heavy Precipitation days	R20	Annual count of days with precipitation larger than 20 mm	day
Max Tmax	TXX	Annual maximum value of daily maximum temperature	°C
Min Tmin	TNN	Annual minimum value of daily minimum temperature	°C
Warm days	TX90P	Percentage of days with Tmax larger than the 90% percentile	%

**Table 3 ijerph-19-06398-t003:** Performance of extreme climate indices for CMIP6 climate models and SD correction.

Index	*R*		$\hat{\sigma }$		*TS*	
	CMIP6	Downscaling	CMIP6	Downscaling	CMIP6	Downscaling
PTOT	0.86 (0.72–0.90)	≈1.00 (≈1.00)	1.30 (1.14–1.75)	1.01 (0.95–1.08)	0.81 (0.64–0.80)	≈1.00 (≈1.00)
RX5	0.88 (0.74–0.94)	0.99 (0.96–0.99)	1.34 (1.03–2.06)	1.00 (0.81–1.20)	0.81 (0.42–0.82)	0.99 (0.93–0.98)
R20	0.79 (0.38–0.88)	0.99 (0.96–0.99)	1.24 (0.93–1.74)	0.81 (0.56–0.98)	0.76 (0.47–0.74)	0.95 (0.78–0.99)
TXX	0.94 (0.78–0.95)	≈1.00 (0.99–1.00)	0.92 (0.95–1.34)	0.99 (0.97–1.06)	0.94 (0.73–0.95)	≈1.00 (≈1.00)
TNN	0.95 (0.84–0.97)	≈1.00 (1.00)	1.79 (1.67–2.24)	1.00 (0.98–1.03)	0.69 (0.47–0.80)	≈1.00 (≈1.00)
TX90P	0.47 (−0.35–0.57)	0.52 (−0.4–0.57)	2.02 (1.45–3.69)	0.59 (0.68–3.26)	0.34 (0.04–0.48)	0.44 (0.09–0.6)

**Table 4 ijerph-19-06398-t004:** The trend coefficients of extreme climate indices over China for observation, MME of CMIP6 and downscaling (/10a.

Dataset	PTOT (%/10a)	RX5 (%/10a)	R20 (d/10a)	TXX (°C/10a)	TNN (°C/10a)	T90P (%/10a)
Observation	0.6	0.3	≈0	−0.1	0.5	1.8
CMIP6	0.09	0.7	≈0	0.2	0.2	2.2
Downscaled	−0.2	0.6	≈0	0.1	0.2	1.6

## Data Availability

Not applicable.

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
