# Peer review of "Simulation and Projection of Climate Extremes in China by a Set of Statistical Downscaled Data"

_ijerph, 2022, doi:10.3390/ijerph19116398_

Round 1
Reviewer 1 Report
The topic is significant and up-to-date.
The abstract must be revised ensuring to be understandable on its own.
The introduction is sufficient.
The methodology is adequate and up-to-date.
The results are very detailed. The analysis of the results should be in the discussion section which is very poor. Some figures must be improved (see the file attached).
The numbering of the sections is incorrect.
The counclusions are specific, but instead of the statements, they shloud highlight that they are predictions based on the model. The discussion part must be improved emphasizing some impacts of the projected changes. Otherwise the results have just methodological soundness but no implications that can be utilized by people who want to accommodate to the predicted climate changes in any sector.
The discussion part must be improved - even it can be integrated in the Results section providing comparisons with the cited references.
Minor corrections are needed in the English grammar of the text.
My comments and corrections are in sticky notes in the pdf file of the manuscript (see attached).

Author Response
Dear Editor,
Thank you for the help and the insightful comments from the three reviewers on our manuscript “Simulation and Projection of Climate Extremes in China by a Set of Statistical Downscaled Data” (ijerph-1704415). The three reviewers all made many helpful comments and suggestions, and we thank this for them. For the revision, we fully considered all suggestions from the three reviewers. We have rewritten the abstract, projection and conclusion parts. Here is the item-by-item reply to the comments. The text with italicization indicates the reviewer’s comments, and the normal text is our response. The revision to the manuscript is highlighted in blue.
Reviewer 1:
The topic is significant and up-to-date. The methodology is adequate and up-to-date. The introduction is sufficient.
- The abstract must be revised ensuring to be understandable on its own.
Response: Thank you for the suggestion. We have rewritten the abstract.
- The results are very detailed. The analysis of the results should be in the discussion section which is very poor. Some figures must be improved (see the file attached).
Response: Thank you for the suggestion. We have rewritten the conclusion and discussion, corrected the errors, and added captions of figures and units in the revised manuscript.
- The numbering of the sections is incorrect.
Response: Sorry for the problem. We have corrected the numbering of the sections in the revised manuscript. Wrong numbers May be caused by reformatting the paper after submission.
- The conclusions are specific, but instead of the statements, they should highlight that they are predictions based on the model. The discussion part must be improved emphasizing some impacts of the projected changes. Otherwise the results have just methodological soundness but no implications that can be utilized by people who want to accommodate to the predicted climate changes in any sector.
Response: Thank you for the suggestion. We have rewritten the results, conclusion and discussion in lines 338-401.
- The discussion part must be improved - even it can be integrated in the Results section providing comparisons with the cited references.
Response: Thank you for the suggestion. We have rewritten results and discussion.
- Minor corrections are needed in the English grammar of the text.
Response: Thank you for the suggestion. We have checked the English grammar of the text, and corrected them.
- My comments and corrections are in sticky notes in the pdf file of the manuscript (see attached).
Response: Thank you for the suggestion and careful correction. We have followed the hints to correct every error.
Reviewer 2 Report
The main aim of this paper was the investigation of the performances of a set of CMIP6 statistical downscaled data in reproducing climate extremes over China. Although the topic is very interesting and worth of investigation, I regret to inform that I cannot recommend the publication, being the manuscript affected by serious issues.
The paper is not well written, there are too many trivial errors, for example: the numbering of sections and subsections is wrong; the link at line 74 does not work; at lines 81-82 the authors write that BCSD is better, but that is affected by large biases in the simulations of extreme events and this seems to me a contradiction.
In the introduction, the authors claim that dynamical downscaling has only disadvantages, but this is not true. Dynamical downscaling is characterized by several advantages with respect to the statistical one, for example it ensures that the variables are physically consistent, it offers a wide range of applications and provides detailed information on climate extremes. In fact, the scientific community is running several initiatives (e.g. CORDEX, CORDEX-CORE) based on regional climate modelling.
The description of the methodology used to for the downscaling is scarce and must be extended, while the description of emission scenarios is completely missing.
The ETCCDI has defined 27 core indices for the description of extreme events, so it is not clear why in this work only 6 have been considered. In a future version, I recommend to consider a larger number of indicators.
Spatial pattern evaluation has been performed considering annual means, but this is not the proper way to proceed, since bias compensation would affect the results. A more appropriate way to proceed is to consider seasonal means (DJF, MAM, JJA, SON).
The authors claim that the downscaling reduces the bias and probably this is true, but plotting only spatial patterns of extreme indices (Figures 1 and 2) makes this analysis quite difficult. It would be better to plot the maps of observations and the maps of the bias (differences between model and observations). In any case, it is evident that some areas are affected by large biases, but the origin of these biases are not investigated.
Temporal trends evaluation (Figure 3) has been performed by averaging data over the whole China, but in this way bias compensation would affect the results, even because the climate features of China vary spatially in a relevant way. It would be better to evaluate temporal trends over smaller subdomains.
Analysis of climate projections is very poor. The consistency of projected spatial patterns of changes with other projections available in literature has not been verified, so nothing can be said about the reliability of these projections. The only comparison mentioned (line 372) is with a work performed by the same authors, moreover it is still under review. There are plenty of projections available in literature, even performed with regional climate modelling (e.g. CORDEX initiative), which can be used as reference.
Author Response
Reviewer 2:
The main aim of this paper was the investigation of the performances of a set of CMIP6 statistical downscaled data in reproducing climate extremes over China. Although the topic is very interesting and worth of investigation, I regret to inform that I cannot recommend the publication, being the manuscript affected by serious issues.
- The paper is not well written, there are too many trivial errors, for example: the numbering of sections and subsections is wrong; the link at line 74 does not work; at lines 81-82 the authors write that BCSD is better, but that is affected by large biases in the simulations of extreme events and this seems to me a contradiction.
Response: Thank you for the suggestion. The wrong numbers of sections and subsections may be caused by changing the format of the paper after submission. We have corrected the numbering of sections. Besides, we have changed the available link at line 74. The statement at lines 81-82 means that the BCSD method can simulate the extreme precipitation indices better than other three methods, but there are some shortcomings in the simulation of the extreme temperature indices.
- In the introduction, the authors claim that dynamical downscaling has only disadvantages, but this is not true. Dynamical downscaling is characterized by several advantages with respect to the statistical one, for example it ensures that the variables are physically consistent, it offers a wide range of applications and provides detailed information on climate extremes. In fact, the scientific community is running several initiatives (e.g. CORDEX, CORDEX-CORE) based on regional climate modelling.
Response: Thank you for the advice. The dynamical downscaling method does have many advantages. We have added the corresponding description “Dynamic downscaling can better simulate the finer-scale physical process such as effects of topography, land-sea distribution and land surface processes, so it can improve the simulation of temperature and precipitation at the regional scale relative to the global model. But it requires high computing resources and is time-consuming” in lines 65-73.
- The description of the methodology used to for the downscaling is scarce and must be extended, while the description of emission scenarios is completely missing.
Response: Thank you for the advice. We have added a description of the statistical downscaling method in lines 129-156. The description of emission scenarios are added in lines 119-125.
- The ETCCDI has defined 27 core indices for the description of extreme events, so it is not clear why in this work only 6 have been considered. In a future version, I recommend to consider a larger number of indicators.
Response: Thank you for your suggestion. The reason why these indices are selected has been added in the revised manuscript in lines 161-163. We do have plans to use more extreme climate indices for future projections. This paper is only an initial attempt at this dataset. We will analyze more extreme indices in the next paper.
- Spatial pattern evaluation has been performed considering annual means, but this is not the proper way to proceed, since bias compensation would affect the results. A more appropriate way to proceed is to consider seasonal means (DJF, MAM, JJA, SON).
Response: Thank you for the suggestion. This article focuses on the analysis of extreme climate indices, which are defined on annual values. We think this is a good proposal. We will explore characteristics at seasonal scale of the indices as well as other indices in the next article.
- The authors claim that the downscaling reduces the bias and probably this is true, but plotting only spatial patterns of extreme indices (Figures 1 and 2) makes this analysis quite difficult. It would be better to plot the maps of observations and the maps of the bias (differences between model and observations). In any case, it is evident that some areas are affected by large biases, but the origin of these biases are not investigated.
Response: Thank you for the suggestion. We mainly want to explore whether the downscaled multi-model ensemble improves the spatial distribution pattern of the extreme climate indices. The results of the quantitative analysis in Table 3 confirm that the downscaling method does improve the simulation of the spatial patterns of extreme climate events. Zhou et al., 2018 and Yang et al., 2019 compare the capability of simulation of spatial patterns only and no bias plots are given. The biases at regional scales and the sources of bias will be further analyzed in next papers.
Li ZHOU, Mingcai LAN, Ronghui CAI, Ping WEN, Rong YAO, Yunyun YANG. Projection and uncertainties of extreme precipitation over the Yangtze River valley in the early 21st century[J]. Acta Meteorologica Sinica, 2018, 76(1): 47-61. doi: 10.11676/qxxb2017.084
Yang Y , Tang J , Xiong Z , et al. An intercomparison of multiple statistical downscaling methods for daily precipitation and temperature over China: present climate evaluations[J]. Climate Dynamics, 2019, 53(5).
- Temporal trends evaluation (Figure 3) has been performed by averaging data over the whole China, but in this way bias compensation would affect the results, even because the climate features of China vary spatially in a relevant way. It would be better to evaluate temporal trends over smaller subdomains.
Response: Thank you for the suggestion. This is a very good advice. In the next paper we will divide China into several sub-regions and evaluate their time-varying characteristics.
- Analysis of climate projections is very poor. The consistency of projected spatial patterns of changes with other projections available in literature has not been verified, so nothing can be said about the reliability of these projections. The only comparison mentioned (line 372) is with a work performed by the same authors, moreover it is still under review. There are plenty of projections available in literature, even performed with regional climate modelling (e.g. CORDEX initiative), which can be used as reference.
Response: Thank you for the suggestion. We have added other papers which evaluate CMIP6 simulations and projections in lines 292-293, 304-307, 330-335.
Reviewer 3 Report
Overall, the result of the manuscript is detailed and the figures are beautiful. However, the manuscript still has some problems and needs to be revised.
1. Line92-97 Please clarify the scientific questions of this study, or the description of the results should focus on the scientific questions raised by this study, rather than covering everything.
2. Line153 Please refine or simplify the result part. Because it looks more like a technical paper than a scientific paper.
3. The serial numbers of the chapter titles in the whole paper are wrong and all are confused. And it happened a lot. Please revise it carefully.
4. Line-341-370 Conclusions and discussion should be rewritten. Please distinguish the conclusion from the discussion section. Usually, the discussion section is used to explain the results, clarify the research point, and indicate the significance of the research. Without it, the study feels more like a report than a scientific paper.
5. Under SSP5-8.5 and SSP2-4.5, extreme climate events are both consistent and different. In general, the frequency and intensity of extreme weather events will be greater in SSP5-8.5. What kind of thinking can this bring? Or why did this study compare the results of extreme climate events under the two scenarios? This is what I expect to see.
Author Response
Reviewer 3:
Overall, the result of the manuscript is detailed and the figures are beautiful. However, the manuscript still has some problems and needs to be revised.
- Line92-97 Please clarify the scientific questions of this study, or the description of the results should focus on the scientific questions raised by this study, rather than covering everything.
Response: Thank you for the suggestion. We have clarified the questions of this study in lines 105-109.
- Line153 Please refine or simplify the result part. Because it looks more like a technical paper than a scientific paper.
Response: Thank you for the suggestion. We have rewritten the part of results
- The serial numbers of the chapter titles in the whole paper are wrong and all are confused. And it happened a lot. Please revise it carefully.
Response: Sorry for the problem, which may be caused by changing format of the paper after submission. We have corrected the numbering of sections.
- Line-341-370 Conclusions and discussion should be rewritten. Please distinguish the conclusion from the discussion section. Usually, the discussion section is used to explain the results, clarify the research point, and indicate the significance of the research. Without it, the study feels more like a report than a scientific paper.
Response: Thank you for the suggestion. We have rewritten conclusions and discussion.
- Under SSP5-8.5 and SSP2-4.5, extreme climate events are both consistent and different. In general, the frequency and intensity of extreme weather events will be greater in SSP5-8.5. What kind of thinking can this bring? Or why did this study compare the results of extreme climate events under the two scenarios? This is what I expect to see.
Response: SSP2-4.5 represents the most likely future emissions scenario, and SSP5-8.5 represents the most extreme scenario with no control of fossil fuel emissions. The comparison under these two scenarios will help to highlight the importance of carbon neutrality. We have add the explanation in lines 123-125.
The reason for more pronounced extreme heavy precipitation in high emission scenario is caused by the thermodynamic responses with more increased moisture availability, while partly offset by the dynamic response to weakened atmospheric circulation (Chen, et al., 2020). The change in vertical circulation and moisture advection is correlated with the change in frequency and intensity of extreme events (Freychet, et al, 2015). The specific humidity of the atmosphere under SSP5-8.5 would increase more than that under SSP2-4.5 in China, so is the vertical wind speed. While the increased intensity of extreme temperature events under higher emission scenarios is easily understood.
References:
Chen, Z., Zhou, T., Zhang, L., Chen, X., Zhang, W., & Jiang, J. (2020). Global land monsoon precipitation changes in CMIP6 projections. Geophysical Research Letters, 47, e2019GL086902. https://doi.org/10.1029/2019GL086902
Freychet, N. , Hsu, H. H. , Chou, C. , & Wu, C. H. . (2015). Asian Summer Monsoon in CMIP5 Projections: A Link between the Change in Extreme Precipitation and Monsoon Dynamics. Journal of Climate, DOI: https://doi.org/10.1175/JCLI-D-14-00449.1